**PLOS** NEGLECTED TROPICAL DISEASES

# Immunoprofiling of fresh HAM/TSP blood samples shows altered innate cell responsiveness

**Brenda Rocamonde**[1ᴼ], **Nicolas Futsch**[1ᴼ], **Noemia Orii** [2], **Omran Allatif** [3], **Augusto Cesar Penalva de Oliveira**[4], **Renaud Mahieux**[1†], **Jorge Casseb**[2], **Hélène Dutartre**[1] *

**1** International Center for Research in Infectiology, Retroviral Oncogenesis Laboratory, INSERM U1111—Université Claude Bernard Lyon 1, CNRS, Ecole Normale Supérieure de Lyon, Université Lyon, Lyon, France, Equipe labelisée par la Fondation pour la Recherche Médicale, Labex Ecofect, **2** Faculdade de Medicina/Instituto de Medicina Tropical de São Paulo/Universidade da São Paulo, São Paulo, SP, Brazil, **3** International Center for Research in Infectiology, service BIBS, INSERM U1111—Université Claude Bernard Lyon 1, CNRS, Ecole Normale Supérieure de Lyon, Université Lyon, Lyon, France, **4** Instituto de Infectologia "Emilio Ribas", São Paulo, SP, Brazil

ᴼ These authors contributed equally to this work.
† Deceased.
* helene.dutartre@ens-lyon.fr

**Data Availability Statement:** All relevant data are within the manuscript and its Supporting Information files.

## Abstract

The Human T-cell Leukemia Virus-1 (HTLV-1)-Associated Myelopathy/Tropical Spastic Paraparesis (HAM/TSP) is a devastating neurodegenerative disease with no effective treatment, which affects an increasing number of people in Brazil. Immune cells from the adaptive compartment are involved in disease manifestation but whether innate cell functions participate in disease occurrence has not been evaluated. In this study, we analyzed innate cell responses at steady state and after blood cell stimulation using an agonist of the toll-like receptor (TLR)7/8-signaling pathway in blood samples from HTLV-1-infected volunteers, including asymptomatic carriers and HAM/TSP patients. We observed a lower response of IFNα⁺ DCs and monocytes in HAM/TSP compared to asymptomatic carriers, as a potential consequence of corticosteroid treatments. In contrast, a higher frequency of monocytes producing MIP-1α and pDC producing IL-12 was detected in HAM/TSP blood samples, together with higher IFNγ responsiveness of NK cells, suggesting an increased sensitivity to inflammatory response in HAM/TSP patients compared to asymptomatic carriers. This sustained inflammatory responsiveness could be linked or be at the origin of the neuroinflammatory status in HAM/TSP patients. Therefore, the mechanism underlying this dysregulations could shed light onto the origins of HAM/TSP disease.

## Author summary

The infection by the Human T-cell Leukemia Virus-1 (HTLV-1) is quite frequent in Brazil. Between 1–5% of infected individuals develop a devastating neurodegenerative disease (HAM/TSP) as a result of a sustained inflammation in the central nervous system, with no effective treatment. So far, inflammation has been linked to the deregulated activation of

**Funding:** This work was supported by University of Sao Paulo and University of Lyon (Fapesp 2014/22827-7 joint program 2015, grant to HD and JC; IDEX-INT-2020-36 to HD and RGE20009CCA-GC20009-CC Joint program 2020 to JC), by Ministério da Saúde do Brasil; Fundação Faculdade de Medicina and Conselho Nacional de Pesquisa Tecnológico (CNP, Grant 301275/2019-0 to JC) by Ligue contre le cancer (Equipe labelisée program EL2013-3Mahieux to RM and HD) and Fondation pour la Recherche Médicale (programme Equipe labelisée, program DEQ20180339200. Grant to RM and HD). NF acknowledge La ligue contre le Cancer for the sponsoring of his PhD fellowship (2015-2018). BR is supported by FRM, HD and OA are supported by INSERM, RM was supported by ENS. The funders had no role in study design, data collection and analysis, decision to publish, or preparation of the manuscript.

**Competing interests:** The authors have declared that no competing interests exist. Author Renaud Mahieux was unable to confirm their authorship contributions. On their behalf, the corresponding author has reported their contributions to the best of their knowledge.

T-cells, but the role of innate cells has not been investigated yet. In this work, we aimed to characterize the responsiveness of innate cells, as this immune population is cornerstone of efficient immune response, but also might participate in disease exacerbation found in chronic infection. Our findings suggest an impaired antiviral response and increased inflammatory responsiveness by dendritic cells and monocytes in HAM/TSP patients compared to asymptomatic carriers. This sustained inflammatory responsiveness upon innate cell activation could participate in the establishment of the HAM/TSP disease.

## Introduction

The Human T-cell Leukemia Virus-1 (HTLV-1)-Associated Myelopathy/Tropical Spastic Paraparesis (HAM/TSP) [1,2] is a progressive neurodegenerative disease characterized by the demyelination of the middle-to-lower thoracic cord [1], with a high prevalence in Brazil [3]. Even though most HTLV-1 infected individuals remain asymptomatic lifelong, around 1–5% develop HAM/TSP. However, anticipating those infected asymptomatic carriers who will develop HAM/TSP remains a challenging task. Viral and immune characterization of HAM/TSP patients identify some markers of pathogenicity, such as an increased proviral load (PVL, *i.e.* the number of integrated copies of the viral genome in the host cells) or higher frequency of CD8$^+$ T-cells and higher secretion of pro-inflammatory cytokines such as TNFα and IFNγ [4]. However, those markers do not allow to anticipate the development of HAM/TSP in infected asymptomatic carriers, since some of them also present an elevated PVL without developing any myelopathy, and alterations in T-cell frequencies and cytokine secretion is only detectable in patients with disease manifestations, and not in infected asymptomatic carriers [5]. Thus, the identification of immune markers predicting the disease progression is required.

HTLV-1 targets mainly T-cells [6], altering their function and ability to induce an antiviral specific immune response, even participating in the disease evolution as mentioned previously. However, in addition to T-cells, HTLV-1 also targets innate immune cells such as classical and plasmacytoid dendritic cells (cDCs and pDCs, respectively) [7–9] as well as monocytes [10]. Yet, their role in the disease manifestation is poorly understood, besides alterations in innate cell frequencies [8,10–12] observed in HAM/TSP patients. Indeed, the frequencies of pDCs [13] were found lower while those of myeloid DCs [13] and of intermediate monocytes [12] were found higher in HAM/TSP patients compared to HTLV-1 asymptomatic carriers and healthy donors. In contrast, while higher frequencies of non-classical monocytes and lower frequencies of classical monocyte were reported in HTLV-1-infected individuals compared to healthy donors [10], these frequencies were not different between infected asymptomatic carriers and HAM/TSP patients [10]. Strikingly, the innate cells responsiveness has not been addressed yet, although a dysfunctional immune response linked either to the infection of innate cells or to their activation upon virus sensing, might be an underlying mechanism involved in HAM/TSP progression. Therefore, innate immune responsiveness and its potential deregulation after HTLV-1 infection require special attention to understand HAM/TSP pathology and disease progression.

The aim of this study was to investigate potential deregulations in innate cell responsiveness in HTLV-1 infected subjects that could indicate a progression towards HAM/TSP. We performed single cell immunoprofiling of freshly collected blood samples from a cohort of asymptomatic carriers and HAM/TSP patients to characterize the phenotype and responsiveness of innate cell subsets after *ex vivo* TLR7/8-stimulation, a broad way to activate most of innate

cells [14] and because, impairment of TLR7/8 signaling upon several chronic infections has been linked to diseases [15,16]. HAM/TSP patients presented a lower frequency of DCs and monocytes producing IFNα upon stimulation compared to asymptomatic carriers, as a potential consequence of corticosteroid treatments expected to reduce inflammatory symptoms of HAM/TSP patients. However, we observed a higher frequency of monocytes and pDC producing IL-12 in HAM/TSP blood samples, together with higher IFNγ responsiveness of NK cells, suggesting a hyper-stimulated inflammatory response in HAM/TSP patients compared to asymptomatic carriers. Altogether, our results highlight for the first time a method to monitor innate cells responsiveness in HTLV-1 infected subjects.

## Methods

### Ethics statement

All individuals included in this study were followed at Institute of Infectious Diseases "Emílio Ribas" (IIER) and signed an informed consent that was approved by the Ethical Board of the Institute of Infectious Diseases "Emílio Ribas" (protocol number 86379218.6.1001.0061).

### Clinical samples

A Brazilian cohort (15 HAM/TSP patients, 15 HTLV-1 asymptomatic carriers, and 15 non-infected individuals) was analyzed. Patients underwent a neurological assessment by a neurologist blinded to their HTLV status. Patients with at least two pyramidal signs, such as paresis, spasticity, hyperreflexia, clonus, diminished or absent superficial reflexes, or the presence of pathologic reflexes (e.g. Babinski sign), were diagnosed for HAM/TSP following De Castro-Costa *et al.* [17] criteria and received corticosteroid treatment (IV methylprednisolone) 45 days apart. Samples were collected at least 15 days before or after this pulse therapy. Asymptomatic carriers were included based on their HTLV-1 positive status and their lack of any HTLV-1 associated clinical symptoms. They were aged and sex matched with the HAM/TSP patients enrolled. Detailed clinical information from HTLV-1 infected individuals included in the cohort is provided in S1 Table.

### HTLV-1 serologic test and proviral load (PVL) determination

HTLV-1 serologic diagnosis was made by ELISA (Ortho Diagnostics, USA) and positive samples were confirmed by western blot (HTLV Blot 2.4 test, DBL, Singapore). All patients whose serum sample was reactive with either test was submitted to a nested-PCR using HTLV-1 generic primers and amplified products were digested with restriction enzymes [18]. In order to determine HTLV-1 proviral load, peripheral blood mononuclear cells (PBMC) were isolated from an acid-citrate-dextrose solution and separated by Ficoll density gradient centrifugation (Pharmacia, Uppsala, Sweden). DNA was extracted using a commercial kit (GFX Pharmacia, Uppsala, Sweden). The forward and reverse primers used for HTLV-1 DNA quantitation were SK110 (5'-CCCTACAATCCAACCAGCTCAG-3', HTLV-1 nucleotide 4758–4779 (GenBank accession No. J02029)), and SK111 (5'-GTGGTGAAGCTGCCATCGGGTTTT-3', HTLV-1 nucleotide 4943–4920). The internal HTLV-1 TaqMan probe (5'-CTTTAC TGACAAACCC GACCTACCCATGGA-3') was selected using the Oligo (version 4, National Biosciences, Plymouth, MI, USA) and Primer Express (Perkin-Elmer Applied Biosystems, Boston, MA, USA) software programs. The probe was located between positions 4829 and 4858 of the HTLV-1 genome and carried a 5' reporter dye FAM (6-carboxy fluorescein) and a 3' quencher dye TAMRA (6-carboxy tetramethyl rhodamine). Albumin DNA quantification was used to normalize variations due to differences of DNA extraction or PBMCs counts as described

previously [19]. The normalized value of HTLV-1 proviral load was calculated as the ratio of (HTLV-1 DNA average copy number/albumin DNA average copy number) x 2 x $10^5$ and is reported as the number of HTLV-1 copies/$10^5$ PBMC [20].

## Whole blood stimulation

Collected blood samples were distributed in 1.5 mL polypropylene tubes and supplemented with 200 μL of RPMI medium containing 10% of fetal bovine serum (FBS). Samples were cultured in the presence of Resiquimod (R848, 1 μg/mL, Invivogen) to simulate TLR7 signaling pathway. Samples cultured in absence of stimulus were used as controls. After 1h of incubation at 37˚C 5% $CO_2$, Brefeldine A (10 μg/mL, Sigma) was added to repress cytokine release. Four hours later, samples were incubated for 10 minutes with ammonium chloride in order to perform the lysis of red cells. Staining was performed after one wash with DPBS 1x (Gibco) on whole leukocytes.

## Phenotypic characterization

Samples were incubated with a Live Dead Aqua Blue reagent (Thermo Fisher Scientific) according to manufacturer instructions. After one wash, cells were saturated with 1% BSA-FcR Blocking (Miltenyi) in DPBS for 15 min at 4˚C, and then surface-stained for 20 min at 4˚C with a cocktail of coupled-antibodies (S2A Table). Leukocytes were then fixed for 20 min at room temperature with 4% paraformaldehyde, permeabilized with 0.05% Saponine-DPBS, and stained with coupled-antibodies directed against intracellular cytokines (S2B Table). Samples were finally analyzed with a LSR Fortessa X-20 cytometer (BD Bioscience). Fluorochrome compensation was performed with compensation beads (BD Bioscience) and FMO (Fluorescence Minus One) conditions.

## Gating strategy

All data were analyzed using FlowJo Software Version 10.5.3 for Mac OS X. Major lineage subsets were selected from forward and side scatter properties followed by single live cells (S1 Fig). Doublet discrimination was achieved by plotting FSC-H vs. SSC-A. For innate immunity live, single, HLA-DR+ cells were selected. Hierarchical gating allows then the discrimination of the following innate cell subsets: cDC1 (CD11c$^+$, BDCA2/3$^+$), cDC2 (CD11c$^+$, BDCA2/3$^-$, BDCA1$^+$), pDC (CD11c$^-$, BDCA2/3$^+$), monocytes (CD11c+, BDCA2/3$^-$, BDCA1$^-$). CD16 and CD14 expression further defined the following subsets of monocytes: classical (CD14$^+$CD16$^-$), intermediate (CD14$^+$CD16$^+$) and non-classical monocytes (CD14$^-$CD16$^+$). NK populations were defined as HLA-DR$^-$, Lin$^-$, and subdivided into CD56$^{dim}$CD16$^+$ and CD56$^{high}$CD16$^-$ NK cell subsets. For the analysis of cytokine production, positive populations were determined after gating determined using fluorescence Minus One (FMO), and the complete gating from innate cell responsiveness of one representative sample is shown in S2 Fig. Gating was applied to all samples and was manually checked for consistency across all samples.

## Biostatistical and computational analyses

(i) **Biostatistical analysis.**   Biostatistical analysis and data processing were performed using the R programming language. In order to determine statistically significant differences between clinical groups, a test on the homogeneity of variances across samples was applied first (Bartlett's test). One-way ANOVA was performed when H0 (= Equal variances) was not rejected, followed by Turkey post-hoc test. Otherwise non-parametric ANOVA (*i.e.* Kruskal-

Wallis test) was applied. Of note, logarithmic transformation of factors not following a normal distribution did not improve statistical performances.

**(ii) tSNE analysis.** T-distributed stochastic neighborhood embedding (tSNE) analysis was performed with FlowJo Software Version 10.5.3 for Mac OS X. cDC11$^+$ and BDCA2-3$^+$ cells were selected for the analysis and a down-sampling of 1,000 cells per samples for each group (i.e. 15,000 cells per group) was performed to have the same number of cells per subject. Only surface markers were selected to perform tSNE analysis with 1,000 iterations and a perplexity of 20. t-SNE analysis included all patients and was presented as a pool for each clinical group.

## Results

### Sociodemographic factors, PVL or innate cell frequencies are similar in AC and HAM/TSP

A total of 45 age-matched individuals were enrolled in the study. In order to match the reported higher prevalence of HTLV-1 infection among women [21], we included three times more women than men. The age, sex, PVL, clinical motor score are detailed in S1 Table. The average age was 52 years (±6.35) for men and 49.15 years (±10.84) for women. Age means were 49.8 years for healthy donors (HD), 46.93 years for asymptomatic carriers (AC) and 53.8 years for HAM/TSP. Although an increased PVL was considered the only hallmark of HAM/TSP [22], we found no significance differences between AC and HAM/TSP subjects (Fig 1A).

We thus wondered whether there was a correlation between the proviral charge and motor dysfunction indicators. No direct correlation was found between the PVL and either IPEC or Osame score of the HAM/TSP patients (Fig 1B). In contrast, the PVL and the age of the patients were correlated (R = 0.45, p-value = 0.033), independently of their clinical status and sex (Fig 1C). This would suggest that high PVL could be the result of either cumulative viral exposure or escape of infected cells from immune surveillance, potentially linked to aged-related immune dysfunctions [23,24], rather than an indicator of the disease onset.

### Dendritic cell responsiveness is impaired in HAM/TSP subjects

Several studies have reported altered cell frequencies and increase of pro-inflammatory cytokines in HAM/TSP subjects compared to asymptomatic carriers [10,12]. However, none of these studies have addressed alterations in cell responsiveness as a potential signature of the disease progression. TLR7/8 signaling is often dysregulated by chronic infection [25]. Aiming at investigating this question, we stimulated the blood collected from HTLV-1-carriers with the TLR7/8 agonist R848. Cells were then immunophenotyped and analyzed by flow cytometry. Dendritic cells were gated from HLA-DR$^+$ subset and classified as cDC1, cDC2 and pDC based on differential expression of CD11c, BDCA2/3 and BDCA1 (see S1 Fig). We found no differences in cDC1 and pDC cell frequencies between the clinical groups and controls. However, we found higher frequencies of cDC2 subset in HAM/TSP patients compared to AC (Fig 2A).

Next, we investigated the cellular heterogeneity of innate cells using an unbiased high-dimensional analysis (tSNE), with the aim to reveal subtle differences in multiple cell populations that may have been missed by the use of biaxial gating. This approach generates a two-dimensional map where similar cells are placed at adjacent points, while cells with different characteristics are separated in space. t-SNE analysis was applied to similar number of 15,000 cells from all individuals in HD, AC and HAM/TSP groups. tSNE analysis showed differences in cDC1 population, particularly in a small population identified in healthy donors and AC

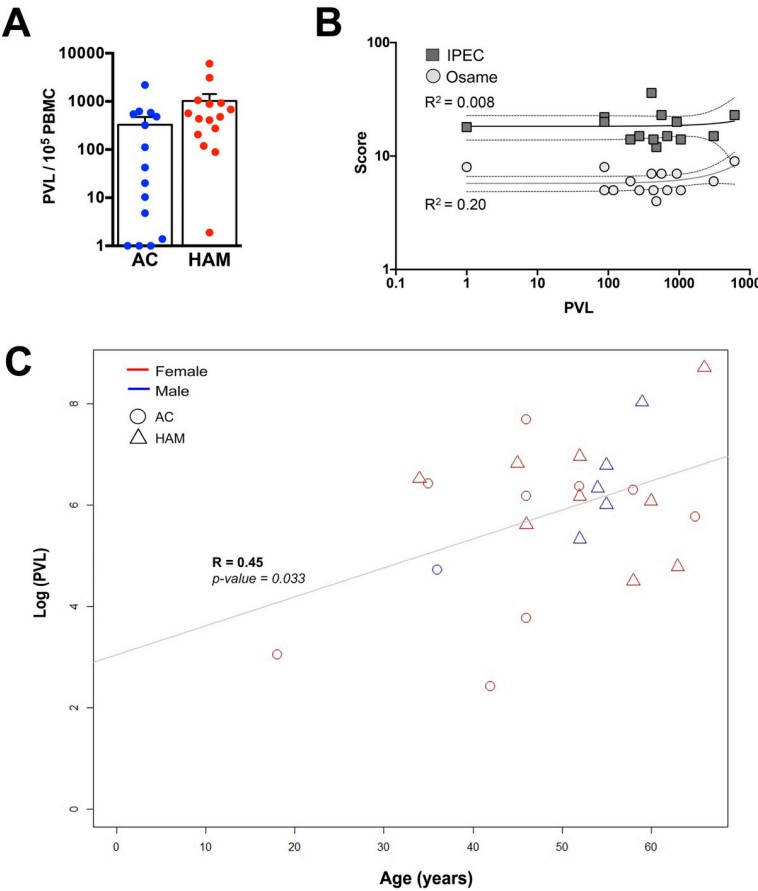

**Fig 1. PVL in PBMCs does not correlate with the clinical status in HTLV-1 infected subjects. (A)** Proviral load (PVL) of the HTLV-1 asymptomatic carriers (AC) and HAM/TSP patients. **(B)** Graph representing motor scores (IPEC and Osame) and PVL in HAM/TSP patients. **(C)** Correlation between PVL of HTLV-1-infected individuals and the age at the time of the analysis.

but not in HAM/TSP patients at steady state (Fig 2B, arrowheads). Interestingly, this small population was maintained after TLR7/8 stimulation only in AC. Then, we investigated the responsiveness of AC and HAM/TSP innate cells by their ability to produce cytokines (IFNα, IL-12, MIP-1α and TNFα) measured at the single cell level (Fig 2C). Overall, TLR7/8 stimulation increased the frequency of cell producing cytokines in all DC subsets from AC and HAM/TSP compared to steady state, suggesting no inhibition of their responsiveness. However, the amplitude of the response was impaired in HAM/TSP samples. Notably, the frequency of cDC1 producing IFNα or TNFα was significantly lower in HAM/TSP samples (Fig 2C upper line), as well as that of cDC2 producing IFNα or MIP-1α (Fig 2C middle line). In contrast, the frequency of pDC producing IL-12 was higher in samples from HAM/TSP patients (Fig 2C lower line). Interestingly, while the frequency of cDC2 producing TNFα, and that of pDC producing TNFα or IFNα were similar in HAM/TSP and AC (Fig 2C middle and lower lines), the median fluorescence intensity (MFI) of these populations was significantly reduced in HAM/TSP patients compared to AC (S4A Fig), strengthening the lower responsiveness.

In order to analyze the quality of DC responsiveness, we analyzed their ability to produce multiple cytokines, using a Boolean analysis at steady state and after TLR7/8 stimulation. Besides lower frequency of TNFα⁺cDC1 in HAM/TSP subjects, we found no significant differences at steady state (Fig 2D and S3A Table). In contrast, the simultaneous co-production of

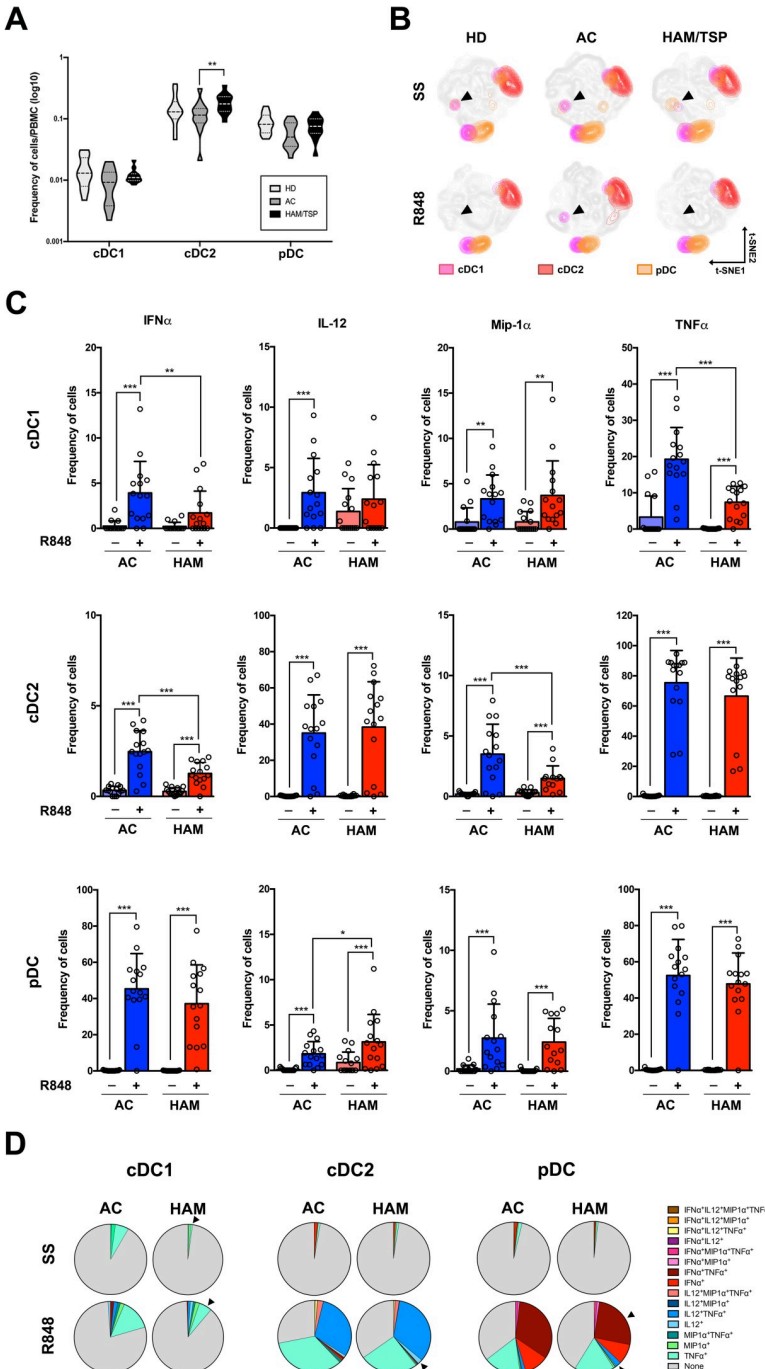

**Fig 2. Dendritic cells of HAM/TSP patients present impaired response to stimulation. (A)** Frequency of dendritic cell subsets (cDC1, cDC2 and pDC) among PBMCs in the clinical groups at steady state. HD: healthy donor; AC: asymptomatic carriers; and HAM/TSP: HTLV-1 associated myelopathy/Tropical spastic paraparesis **(B)** tSNE analysis of DC subsets in innate cell populations at steady state and after TLR7-stimulation with R848. **(C)** Frequency of the cells producing IFNα, IL-12, MIP-1α and TNFα at steady state and after R848 treatment in the different DC subsets. Statistical significance was determined using one-way ANOVA followed by Turkey post-hoc test. * *p-value* ≤ 0.05; ** *p-value* ≤ 0.01; *** *p-value* ≤ 0.001. **(D)** Pie-chart of the Boolean analysis, using the data from all samples of each clinical groups, for the cytokine production in DC subsets at steady state (SS) and after TLR7 stimulation (R848).

cytokines by innate cells in stimulated samples from HAM/TSP patients was impaired (Fig 2D). HAM/TSP patients presented a significant lower frequency of cDC1 producing TNFα alone, a significant lower frequency of MIP-1α$^+$TNFα$^+$-producing cDC2 and a variation of the frequencies of IFNα$^+$ and IFNα$^+$TNFα$^+$-producing pDCs (arrowheads in Fig 2D and S3B Table) compared to AC subjects. Finally, IL-12 producing pDC were slightly higher in HAM/TSP subjects compared to AC group. Altogether, this suggest that production of inflammatory cytokines by dendritic cells from HAM/TSP is lower probably as a consequence of the corticosteroid treatment.

## Monocytes from HAM/TSP patients present greater IL-12 and MIP-1α response to stimulation

Monocytes cell counts and functions are impaired by HTLV-1 infection [10], we thus investigated their responsiveness after TLR7/8 stimulation. Monocytes were gated from HLA-DR$^+$CD11c$^+$ subset excluding BDCA1 expression and subsequently divided in three subpopulations based on the expression of CD16 and CD14 markers as: classical monocytes (cMono; CD14$^+$CD16$^-$), intermediate monocytes (intMono; CD14$^+$CD16$^+$) and non-classical monocytes (ncMono; CD14$^-$CD16$^+$) (Figs 3A and S1). Consistent with a previous report [12], a higher frequency of intermediate monocytes was detected in HAM/TSP patients compared to asymptomatic carriers (Fig 3B).

tSNE distribution revealed subtle differences in the distribution of classical and intermediate monocytes between the three clinical groups (Fig 3C). Two clusters of intermediate monocytes were identified based on t-SNE plots, unequally distributed between AC and HAM/TSP especially after TLR7/8 stimulation.

In contrast to what observed in dendritic cells, we found overall greater responsiveness of HAM/TSP monocytes to TLR7/8 stimulation (Fig 3D). Frequencies of IL-12-producing classical and non-classical monocytes, as well as that of MIP-1α intermediate monocytes were significantly higher in HAM/TSP subjects than in asymptomatic carriers. In contrast, the frequency of MIP-1α classical monocytes was lower in HAM/TSP compared to AC.

Regarding the Boolean analysis, the spontaneous production of TNFα by intermediate and non-classical monocytes was significantly lower in HAM/TSP patients compared to AC (Fig 3E and S3A Table). In contrast, the frequency of classical and intermediate monocytes co-producing IL-12 and TNFα after TLR7/8 stimulation were significantly higher in samples from HAM/TSP patients compared to AC (indicated by an arrowhead in Fig 3E and S3B Table). No significant differences in MFI of any monocyte populations from HAM/TSP or AC was observed (S4B Fig).

Altogether, the signature detected in samples from HAM/TSP patients after stimulation can be summarized as a reduction of proinflammatory cytokines (TNFα and IFNα) combined to an increase of IL-12 produced by different innate subsets.

## HAM/TSP patients present greater IFNγ-producing cells after stimulation

Natural Killer cells link the innate and the adaptive immune response [26], especially in autoimmune disease [27]. Furthermore, NK cell population of HTLV-1 infected subjects showed spontaneous proliferation capacity [28]. In line with this, we aimed at investigating potential frequency alterations in our cohort of HTLV-1 infected donors. NK cells subsets were gated from HLA-DR$^-$ subset and divided into CD56$^{dim}$CD16$^+$ and CD56$^{high}$CD16$^-$ (see S1 Fig). NK cells are recognized as a heterogenous population based on the expression of different receptors. Classically NK cells are divided in CD56$^{dim}$ or CD56$^{high}$ based on their functions: CD56$^{dim}$CD16$^{high}$ NK cell subset expresses high levels of perforin and mediates natural and

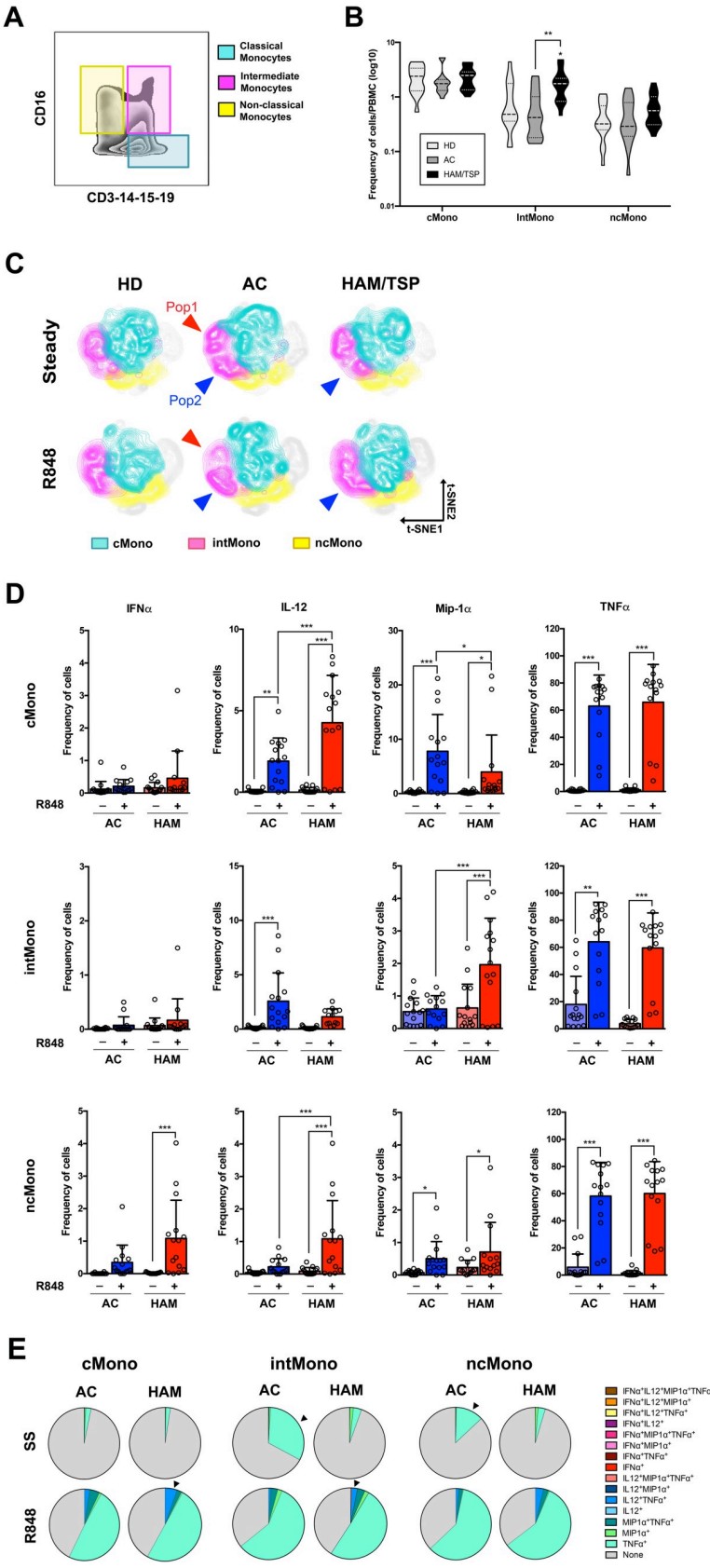

**Fig 3. Monocytes from HAM/TSP patients present greater IL-12 and MIP-1α response to stimulation. (A)** Gating strategy for identification of the three monocyte subpopulations using CD16 and CD3-14-15-19 antibodies. **(B)** Frequency of innate subsets in the clinical groups evaluated in monocytes subsets (classical monocytes, intermediate monocytes and non-classical monocytes). **(C)** tSNE analysis of monocyte subset frequencies in whole blood samples at steady state and after TLR7-stimulation with R848. **(D)** Frequency of the cells producing IFNα, IL-12, MIP-1α and TNFα at steady state and after R848 treatment in the different monocyte subsets. HD: healthy donor; AC: asymptomatic carriers; and HAM/TSP: HTLV-1 associated myelopathy/Tropical spastic paraparesis. Statistical significance was determined using one-way ANOVA followed by Turkey post-hoc test. * *p-value* ≤ 0.05; ** *p-value* ≤ 0.01; *** *p-value* ≤ 0.001. **(D)** Pie-chart of the Boolean analysis for the cytokine production in monocyte subsets at steady state (SS) and after TLR7 stimulation (R848).

antibody-dependent cellular cytotoxicity, and enhanced killing while CD56^high^CD16^±^ NK cells are characterized by low levels of perforin, and are primarily specialized for cytokine production, a function that could be deregulated as a consequence of inflammatory status of HAM/TSP disease.

We observed a higher frequency of both NK cell subsets in asymptomatic carriers compared to healthy donors. In contrast, HAM/TSP patients presented lower cell frequencies of CD56^high^CD16^-^ NK cells compared to asymptomatic carriers (Fig 4A). tSNE distribution evinced an extra subpopulation of CD56^dim^CD16^+^ NK cells in asymptomatic carriers (Pop1)

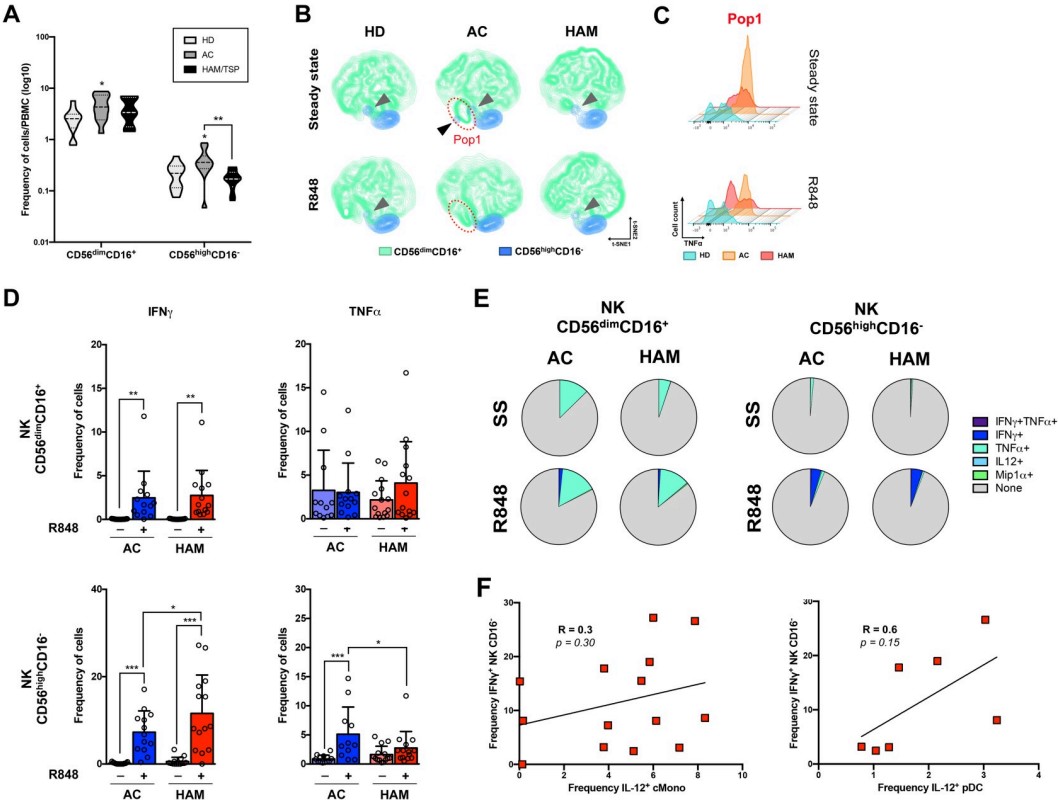

**Fig 4. NK cells from HAM/TSP present greater response to IFNγ after stimulation. (A)** Violing-plot representation of the cell frequency of NK subpopulations in the 3 clinical groups. **(B)** t-SNE clustering of the NK cells reveals different subpopulations and **(C)** the histogram represents TNFα expression of the identified population (Pop1). **(D)** Bar-plot representing the frequency of cytokine producing cells by the gated NK subpopulations at steady state and upon R848 stimulation. **(E)** Boolean analysis of the multiple-cytokine production by the NK subtypes at steady state (NS) and after TLR7/8 stimulation (R848). **(F)** Correlation between the frequency of IFNγ^+^ NK CD16^+^ and the frequency of IL-12^+^ classical monocytes (left) or pDC (right) after stimulation.

at steady state (Fig 4B), presenting high production of TNFα at steady state but not after stimulation (Fig 4C). A small subpopulation of CD56$^{dim}$CD16$^-$ NK cells disappeared in asymptomatic carriers after TLR7/8 stimulation.

The interaction of NK cells with macrophages, lymphocytes and dendritic cells modulates their multifactorial role as cytotoxic effectors and/or protectors in disease progression [27]. Thus, we wondered if the observed alteration in dendritic cells and monocytes responsiveness could trigger an NK dysfunctional response. Both NK cells subset of AC and HAM/TSP responded to stimulation through greater frequency of IFNγ-producing cells compared to steady state. The responsiveness of CD56$^{dim}$CD16$^-$ NK cells in HAM/TSP patients was statistically higher compared to that of AC (Fig 4D). In contrast, the frequency of CD56$^{dim}$CD16$^+$ NK cells producing TNFα was similar in both steady state and after stimulation suggesting a lack of response in both AC and HAM/TSP (Fig 4D, upper panel). However, while CD56$^{dim}$CD16$^-$ NK producing TNFα from AC retained the ability to respond to stimulation those from HAM/TSP did not, resulting in statistical lower frequency of TNFα-producing CD56$^{dim}$CD16$^-$ NK after stimulation in HAM/TSP compared to AC (Fig 4D, lower panel). Boolean analysis didn't show production of multiple cytokines by NK and no difference between AC and HAM/TSP was observed (Fig 4E). IFNγ production by NK cells is enhanced by IL-12 [29], we thus asked whether the increased frequencies of pDC and classical monocytes producing IL-12 we observed in HAM/TSP patients (Figs 2C and 3D respectively), could be correlated to the higher frequency of CD56$^{dim}$CD16$^-$ NK producing IFNγ (Fig 4F). No correlation was observed with IL-12$^+$ classical monocytes (Fig 4F left panel), however the responsiveness of CD56$^{dim}$CD16$^-$ NK producing IFNγ seemed to correlated with frequency of pDC producing IL-12, although it did not reach statistical significance (Fig 4F, right panel).

## Discussion

Despite its low frequency (1–5% of HTLV-1-infected subjects), HAM/TSP represents a devastating neurodegenerative disease with no effective treatment to date. Moreover, prediction of evolution towards HAM/TSP pathology in HTLV-1 asymptomatic carriers is still a challenging task due to the lack of predictive markers. Most of the innate immune analysis performed in HTLV-1 infected subjects have been focused on the evaluation of cell frequency alterations upon viral infection [10,12]. Indeed, previous studies reported conflicting results that, consistent to our work, do not allow to define HAM/TSP predictive markers using cryopreserved blood samples [10,12,13]. Interestingly while, innate cell frequencies are apparently not affected by cryopreservation, we noticed that innate cell responsiveness is lost when blood samples were frozen before analysis (S5 Fig). Here we address for the first time a single-cell immunoprofiling of spontaneous cytokine production and innate cells' responsiveness upon TLR7/8-signaling pathway stimulation in fresh blood samples of HTLV-1-infected subjects. Our first objective was to identify potential immune biomarkers that could sign the disease before symptomatic manifestations. Our second objective was to evaluate immune responsiveness of HAM/TSP patients under treatment at the innate immune functions level.

We provided here that neither an elevated PVL–an average hallmark of HAM/TSP patients compared to asymptomatic carriers [22]–nor immune cell frequency alterations are sufficient to anticipate the disease progression. This might not be in concordance with reports from other groups, although we observed an increased tendency of the mean of PVL in HAM/TSP without reaching statistical significance. Indeed, we observed in our cohort diagnosed HAM/TSP patients with lower PVL compared to that of AC and diagnosed HAM/TSP patients with PVL similar to that of AC. This thus suggests that the absolute PVL is not per se a fair indicator of the disability or disease degree, and highlights the important dispersion of PVL in

both groups. To our knowledge, literature report no information regarding the effect of HAM/TSP treatment on PVL and no longitudinal analysis were performed in our study leaving open the role of corticoid treatment on PVL or motor disability. However, PVL seems to correlate with age potentially as a result of immune viral escape or cumulative viral exposition. Our results suggest an efficiently reduction of innate immune inflammatory responsiveness in HAM/TSP patients upon treatment, specially resulting in reduced production of TNFα and IFNα by dendritic cells compare to that of AC. Nevertheless, no apparent improvement the disease-associated clinical status upon treatment was observed, suggesting potential irreversible CNS damage. Moreover, higher frequency of cDC1 producing TNFα or cDC2 producing IFNα in HAM/TSP seems to correlate with higher clinical IPEC score although without statistical significance (S3A Fig). On the other hand, monocytes from HAM/TSP patients presented higher responsiveness to produce IL-12 and MIP-1α, without any correlation with IPEC score. Together with the higher number of intermediate monocytes, this could lead to a significant increase in the overall production of both IL-12 and MIP-1α in HAM/TSP patients. Interestingly, MIP-1α, also named CCL3, is an inflammatory chemokine associated with multiple sclerosis (MS) [30], an autoimmune and inflammatory disease that affects brain and spinal cord functions, through demyelination of nerves, causing irreversible damages of the CNS. In MS, MIP-1α stimulates T-cell chemotaxis from periphery to inflamed tissues and regulates the trans-endothelial migration of monocytes, dendritic cells and NK cells [31]. Thus, the increased responsiveness of monocytes towards the production of MIP-1α in HAM/TSP individuals despite corticoid treatments could favor neuro-invasion of immune cells into the CNS, maintaining neuroinflammation. Interestingly, potent antagonists of CCL3 receptors, CCR1 and CCR5, have been developed [32] and their efficacy evaluated in clinical trials against multiple sclerosis among other inflammatory diseases. Thus, CCR1 antagonists might also be considered in the treatment of HTLV-1 infected individuals at risk of HAM/TSP.

In addition, the greater responsiveness of some DC subsets and monocytes to produce IL-12 in HAM/TSP patients could contribute to the maintenance of an adaptive inflammatory response, despite anti-inflammatory treatment, as IL-12 strongly synergizes with other stimuli to induce a maximal production of IFNγ [33,34] in T-cells and enhance NK cells cytotoxic activity [35]. Interestingly, the frequency of pDC producing IL-12 seems also to correlate with the IPEC score of HAM/TSP patients, although not reaching statistical significance (S3B Fig). In line with this, we detected higher frequency of IFN-γ producing NK cells in HAM/TSP patients after stimulation, and at steady state, which together is reminiscent to a continuous activation state [36]. Altogether, our results highlight an innate immune signature in HAM/TSP patients different from HTLV-1 carriers with induction of inflammatory cytokines produced by innate cells of HAM/TSP patients despite corticoid treatment.

During the course of the analysis, 5 out of 15 asymptomatic carriers (AC) presented light signs of inflammation that are believed to be the first signs of evolution towards HAM/TSP [37]. However, a retrospective analysis of these subjects failed to identify functional or phenotypic differences in innate cells linked with disease progression. This highlights the important added value of longitudinal studies of HTLV-1-asymptomatic carriers from larger cohorts to identify early immune signs of disease evolution towards HAM/TSP in a significant number of HTLV-1-asymptomatic individuals at risk of severe disease development. Such longitudinal blood analysis would increase the probability to identify predictive markers of disease evolution, which until now are still missing besides several investigations. Finally, our work stresses the need to work with fresh blood samples, unbiased analysis of innate cells' responsiveness and longitudinal samples from large cohort of HTLV-1 carriers to increase the probability of identifying predictive markers of HAM/TSP evolution.

## Supporting information

**S1 Fig. Hierarchical gating strategy of the different immune cell subpopulations.** Flow cytometry collected datasets were analyzed with FlowJo software. A total of $2 \times 10^6$ cells were registered and selected by cell size and granularity. After selection of single cells, viable cells were gated and innate immune cell populations were identified as indicated.
(TIF)

**S2 Fig. Gating strategy for cytokines.** Example of the gating strategy for cytokine determination in AC and HAM/TSP group for IFNα, IL-12 MIP-1α and TNFα in the different cell subsets.
(TIF)

**S3 Fig. Cell responsiveness tends to correlate with clinical score.** Correlation between IPEC Score of HAM/TSP patients and the frequency of (**A**) IFNα+ cDC2 and TNFα+ cDC1; and (**B**) IL-12+ pDC after R848 stimulation. Spearman test was applied to determine the correlation between the two factors.
(TIF)

**S4 Fig. HAM/TSP present lower MFI.** Median intensity fluorescence (MFI) of the cytokine expression for dendritic cell (**A**) and monocytes (**B**) subsets after TLR7 stimulation in AC and HAM/TSP patients. One-way ANOVA followed by Sidak's correction for multiple comparisons was applied.
(TIF)

**S5 Fig. Innate cells from frozen PBMCs lose their responsiveness. A.** A total of $2 \times 10^6$ cells from fresh or frozen PBMCs were registered and selected by cell size and granularity. After selection of single cells, viable cells were gated and innate immune cell populations were identified as indicated in S1 Fig. **B.** PBMCs from fresh or frozen PBMCs were stimulated with R848 and analyzed by flow cytometry for their intracellular production of IL-12 and TNFα (for BDCA3+ cDC1; BDCA1 cDC2 and monocytes) or IFNα and TNFα (for pDC).
(TIF)

**S1 Table. Clinical information of the cohort.** Clinical status, sex, age, PVL, motors score and treatment information is detailed for each HTLV-1-infected subjects enroller in the study.
(TIF)

**S2 Table. List of antibodies.** Recapitulative list of the (**A**) membrane markers antibodies and (**B**) intracellular markers antibodies used for the analysis of the innate immune response by flow cytometry.
(TIF)

**S3 Table. List of cell frequencies in Boolean analysis.** Cell frequency of multi-cytokine production determined using boolean analysis at (A) steady state and (B) after TLR7/8 stimulation.
(TIF)

## Acknowledgments

BR would like to acknowledge Dr. Yamila Rocca (CIRI, T Walzer team). Authors specifically thank Sebastien Dessurgey and Thibault Andrieu (SFR biosciences, cytometry platform) for their technical advices and Dr. Chloé Journo for her critical reading, helpful discussions and her help in the statistics. Authors are also grateful to Dr. David Karlin for his advices in writing and to Dr Patrick Lecine for critical reading. Author Renaud Mahieux passed away during the course of the publication process. Authors would like to dedicate this work to his memory.

## Author Contributions

**Conceptualization:** Augusto Cesar Penalva de Oliveira, Renaud Mahieux, Jorge Casseb, Hélène Dutartre.

**Data curation:** Brenda Rocamonde.

**Formal analysis:** Brenda Rocamonde, Nicolas Futsch, Noemia Orii, Omran Allatif, Hélène Dutartre.

**Funding acquisition:** Renaud Mahieux, Jorge Casseb, Hélène Dutartre.

**Investigation:** Nicolas Futsch, Hélène Dutartre.

**Methodology:** Brenda Rocamonde, Nicolas Futsch, Noemia Orii, Hélène Dutartre.

**Project administration:** Hélène Dutartre.

**Supervision:** Hélène Dutartre.

**Validation:** Brenda Rocamonde, Nicolas Futsch, Renaud Mahieux, Hélène Dutartre.

**Visualization:** Brenda Rocamonde.

**Writing – original draft:** Brenda Rocamonde, Hélène Dutartre.

**Writing – review & editing:** Brenda Rocamonde, Noemia Orii, Renaud Mahieux, Jorge Casseb, Hélène Dutartre.

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
