## [Decision Letter · Decision Letter 0]

24 Aug 2021

Dear Dr. Hélène Dutartre

Thank you very much for submitting your manuscript "Immunoprofiling of fresh HAM/TSP blood samples show altered innate cell responsiveness." for consideration at PLOS Neglected Tropical Diseases. As with all papers reviewed by the journal, your manuscript was reviewed by members of the editorial board and by several independent reviewers. In light of the reviews (below this email), we would like to invite the resubmission of a significantly-revised version that takes into account the reviewers' comments. 

We cannot make any decision about publication until we have seen the revised manuscript and your response to the reviewers' comments. Your revised manuscript is also likely to be sent to reviewers for further evaluation.

Sincerely,

Masao Matsuoka, M.D., Ph.D.

Deputy Editor

Reviewer's Responses to Questions

**Key Review Criteria Required for Acceptance?**

**Methods**

-Are the objectives of the study clearly articulated with a clear testable hypothesis stated?

-Is the study design appropriate to address the stated objectives?

-Is the population clearly described and appropriate for the hypothesis being tested?

-Is the sample size sufficient to ensure adequate power to address the hypothesis being tested?

-Were correct statistical analysis used to support conclusions?

-Are there concerns about ethical or regulatory requirements being met?

Reviewer #1: The methods are satisfactory, with only minor corrections required (see summary and general comments). The study design is appropriate, the population is clearly described and is of appropriate sample size. The statistical analysis seems appropriate and no ethical or regulatory concerns were identified.

Reviewer #2: (No Response)

Reviewer #3: (No Response)

**Results**

-Does the analysis presented match the analysis plan?

-Are the results clearly and completely presented?

-Are the figures (Tables, Images) of sufficient quality for clarity?

Reviewer #1: The results are clearly and completely presented and the figures and tables are of high quality.

Reviewer #2: (No Response)

Reviewer #3: (No Response)

**Conclusions**

-Are the conclusions supported by the data presented?

-Are the limitations of analysis clearly described?

-Do the authors discuss how these data can be helpful to advance our understanding of the topic under study?

-Is public health relevance addressed?

Reviewer #1: The conclusions are supported by the data, and limitations of the study are appropriately identified and dealt with in the discussion.

Reviewer #2: (No Response)

Reviewer #3: (No Response)

**Editorial and Data Presentation Modifications?**

Reviewer #1: The manuscript requires some minor editing for English usage. E.g. ‘seed’ line 36; line 306 ‘make the bond’; line 308 ‘in this line’ etc.

Reviewer #2: (No Response)

Reviewer #3: (No Response)

**Summary and General Comments**

Reviewer #1: Review of “Immunoprofiling of fresh HAM/TSP blood samples show altered innate cell responsiveness”. This manuscript evaluated innate immune responses in fresh blood samples from HAM/TSP patients, asymptomatic carriers of HTLV-1 and healthy controls. The authors stimulated whole blood with a TLR7/8 agonist and evaluated the production of IFN-a, IFN-g, MIP1a, TNF-a and IL-12 by dendritic cells (DCs), monocytes and natural killer (NK) cells by intracellular cytokine staining coupled with detailed immunophenotyping. 

The authors observed that the frequency of DCs, monocyte and NK cell subsets was broadly similar in all study participants. Intermediate monocytes and cDC2 were present at higher frequencies in HAM/TSP patients, and CD56highCD16- NK cells were present at lower frequencies in HAM/TSP patients. There were also several notable differences in cytokine production between ACs and HAM/TSP patients: HAM patients had higher frequencies of IL-12+pDCs, IL-12+cMonocytes, MIp1a+IntMonocytes, IL-12+ncMonocytes and IFN-g+CD56highCD16+ NK cells than ACs, and lower frequencies of IFN-a+cDCs, TNFa+cDCs, IFN-a+cDC2s, MIP-1a+ cDC2s, MIP-1a+cMonocytes and TNFa+CD56highCD16+ NK cells. 

The authors propose that increased production of cytokines in response to TLR7/8 stimulation observed in HAM/TSP patients might be linked to the neuroinflammation in HAM/TSP patients, and the reduced responsiveness of DCs and monocytes could represent incomplete repression of innate immune responses by corticosteroid treatment in HAM patients.

The paper is well written and would be of interest to those studying HTLV-1 pathogenesis, innate immune responses in retroviral infection and inflammatory disease. HAM/TSP is an understudied consequence of HTLV-1 infection, and this study addresses the important question of whether innate immune responses are dysregulated in HAM/TSP patients. Due to the nature of the cohort available (all HAM/TSP patients studied had received corticosteroid treatment), it is impossible to determine whether the observations relate to treatment or disease status. However, the cohort of patients is described comprehensively, experimental techniques and data are clearly presented and appropriately interpreted. Furthermore, the data highlight that evaluation of fresh samples is essential in investigating innate immunopathogenesis in viral infection, and publication of this manuscript will justify future much-needed studies. 

Minor concerns

Monocyte gating data is missing from S fig 1, NK gating differs in fig 4a and S fig 1. NK cytokine gating data missing is from S fig 2.

Do the t-SNE and boolean analyses presented in fig 2/3/4 represent data from an individual patient from each group, or were all patients included? 

The discussion refers to predictive power for future progression, however longitudinal data was not presented. E.g. line 358/359

Similarly, the line ’Sociodemographic factors, PVL or innate cell frequencies do not predict a HAM/TSP progression’ (line 181-182) should be modified to reflect the results presented.

Reviewer #2: Rocamonde, et al. investigated cytokine responses of dendritic cells and monocytes upon stimulation with TLR7/8 stimulation in patients with HAM/TSP and compared them to those of ACs. 

Despite the inflammatory disease manifestation, the frequency of IFNa and MIP1a-producing DCs in HAM/TSP patients was lower than ACs. In contrast, IL-12 and MIP1a responses in the monocytes were higher in HAM/TSP patients. This study demonstrated the different inflammatory responses upon TLR7/8 stimulation between HAM/TSP patients and ACs. However, since all the HAM/TSP patients evaluated were under treatment, it is difficult to judge whether the differences demonstrated could be a consequence of corticosteroid treatment or representing disease-specific response. In addition, the cytokine responses against TLR7/8 stimulation in seronegative controls were not shown. Although this study includes potentially important information, several points should be improved. 

Major points

1. The authors speculated that the reduction of innate immune response in DCs could be a consequence of corticosteroid treatment. Indeed, corticosteroid is known to suppress inflammatory cytokines. However, in this study, they showed discrepancies between IFNa/TNFa and IL-12/IFNg responses and also between the responses of DCs and monocytes. It may be difficult to examine the responses in HAM/TSP without treatment, but the authors should at least assess the cytokine responses to TLR7/8 in DCs and monocytes from seronegative individuals in the presence or absence of corticosteroids in vitro. 

2. The finding of enhanced IL-12 response in HAM/TSP patients is interesting and potentially important. However, it is curious that the authors described this inflammatory cytokine as an anti-inflammatory cytokine (line #302-303) but did not examine a representative anti-inflammatory cytokine IL-10. Is there any data on IL-10 response?

3. Correlation between IPEC score and the frequency of DCs positive for IFNa, TNFa or IL-12 was shown in Suppl. Fig. 3, although it was not statistically significant. As the authors clearly showed enhanced MIP1a and IL-12 responses in HAM/TSP monocytes, correlation between the monocyte responses and IPEC score should be shown also. 

Minor points

1. Suppl. Table 1. Does ‘Treatment’ mean corticosteroid treatment in all the patients shown? It should be clearly described about the use of corticosteroid in the column of ‘Treatment.’

2. Suppl. Table 1. What is the purpose to show ‘breastfeeding’ without information of HTLV-1 infection in mothers and ‘familiar situation’ with marital status? These may be confusing.

Reviewer #3: Rocamonde et al. analyzed innate cells including dendritic cells, monocytes, and NK cells from HAM/TSP patients using fresh blood samples and compared them with those from HTLV-1 carriers. They found some dysregulation of these immune cells from HAM/TSP patients, high frequencies of MIP-1a producing monocytes, IL-12 producing pDC, and IFN-g producing NK cells. The study comprehensively profiled the innate cells in HAM/TSP, however, several issues need to be addressed before publication. 

Major points;

1. In line 203-205, this sentence is an interpretation of the data and should be moved to the Discussion. The authors found no differences in the PVL between HAM/TSP patients and HTLV-1 carriers, which is controversial to the previous reports from many research groups. Please discuss this discrepancy in the Discussion. Is there a possibility that the treatment affected the PVL or motor disability?

2. In line 259, they suspected that the low responsiveness in cytokine is due to the corticosteroid treatment. If so, this study may not reflect the responses of innate immune cells in the natural course of the disease. I strongly recommend that they collect blood samples from untreated HAM/TSP patients and analyze them, especially if they are looking for a predictive immune marker for progress towards HAM/TSP from carrier state.

3. In Figures 2 to 4, they detected cytokine-producing cells by flow cytometry and compared the frequency in each cell population. However, the amount of the produced cytokines from each cell is not clear to be the same, I recommend measuring the amount of cytokines in the culture supernatant by ELISA or use the mean fluorescence intensity reflecting the cytokine amount of the cells. 

4. The authors found that IL-12-producing DC and monocytes were high frequency in HAM/TSP patients. It is known that IL-12 stimulates NK cells. Do the increased IL-12 affect the NK cell function and modify the clinical status of HAM/TSP patients?

Minor points;

1. In line 121, 10 to the 6th power may be 10 to the 5th power.

2. In supporting figure 1 and line 155, the CD56highCD16- cell population contains CD56dimCD16- cell population, but in Figure 4A, the cell population did not. Please correct the supporting figure 1.

3. In line 202, “independently of their clinical status and sex” is inappropriate because the data included all samples but is not divided by these attributes and compared among subgroups.

4. In line 212, did they stimulate only the blood from HTLV-1 carriers?

5. In line 225, MIP-1 alpha is a more general expression than Mip-1 alpha. 

6. In Figure 3A, they defined the classical monocytes as CD16 negative but positive for CD3/14/15/19 cocktail. I think monocytes do not express either CD3 or CD19. Monocytes were firstly gated from HLA-DR+CD11c+, suggesting that this classical monocyte population did not contain T cells and B cells. Why did not the authors use a CD14 antibody instead of the antibody cocktail? 

7. Please explain why the authors divided the NK cells into two subsets, CD56dim and CD56high.

8. In line 394, the increased IFN-g production of NK cells was induced by R848 stimulation in vivo. Why can the authors argue that corticoid treatment failed to control the production of inflammatory cytokines?

PLOS authors have the option to publish the peer review history of their article (what does this mean?). If published, this will include your full peer review and any attached files.

Reviewer #1: No

Reviewer #2: No

Reviewer #3: No
---

## [Decision Letter · Decision Letter 1]

12 Oct 2021

Dear Dr. Helene Dutartre,

Thank you very much for submitting your manuscript "Immunoprofiling of fresh HAM/TSP blood samples show altered innate cell responsiveness." for consideration at PLOS Neglected Tropical Diseases. As with all papers reviewed by the journal, your manuscript was reviewed by members of the editorial board and by three independent reviewers. Based on the reviews, we are likely to accept this manuscript for publication, providing that you modify the manuscript according to the review recommendations. 

Sincerely,

Masao Matsuoka, M.D., Ph.D.

Deputy Editor

Reviewer's Responses to Questions

**Key Review Criteria Required for Acceptance?**

**Methods**

-Are the objectives of the study clearly articulated with a clear testable hypothesis stated?

-Is the study design appropriate to address the stated objectives?

-Is the population clearly described and appropriate for the hypothesis being tested?

-Is the sample size sufficient to ensure adequate power to address the hypothesis being tested?

-Were correct statistical analysis used to support conclusions?

-Are there concerns about ethical or regulatory requirements being met?

Reviewer #1: Methods meet the required standards.

Reviewer #2: (No Response)

Reviewer #3: (No Response)

**Results**

-Does the analysis presented match the analysis plan?

-Are the results clearly and completely presented?

-Are the figures (Tables, Images) of sufficient quality for clarity?

Reviewer #1: Results meet the required standards.

Reviewer #2: (No Response)

Reviewer #3: (No Response)

**Conclusions**

-Are the conclusions supported by the data presented?

-Are the limitations of analysis clearly described?

-Do the authors discuss how these data can be helpful to advance our understanding of the topic under study?

-Is public health relevance addressed?

Reviewer #1: Conclusions are clear and appropriate to the data presented.

Reviewer #2: (No Response)

Reviewer #3: (No Response)

**Editorial and Data Presentation Modifications?**

Reviewer #1: Not required.

Reviewer #2: (No Response)

Reviewer #3: (No Response)

**Summary and General Comments**

Reviewer #1: All of my concerns have been addressed and I recommend publication of the manuscript.

Reviewer #2: In the revised manuscript, the authors made several modifications to improve the clarity. However, they could not add any data about the effect of corticosteroids. It is unfortunate that the clinical significance of the observations remains obscure. Nevertheless, this study contains valuable data of fine immunological analysis using rare clinic samples.

Reviewer #3: Rocamonde et al. responded well to my comments. However, there are still some points to be improved. Followings are new comments corresponding to the previous comment numbers. 

Major points;

1. The authors found no differences in the PVL between HAM/TSP patients and HTLV-1 carriers, which is controversial to the previous reports from many research groups. This point is important. In the point-to-point reply, the authors try to explain why the discrepancy occurs. I recommend writing like this in the Discussion section. 

2. I understand how difficult to collect fresh blood samples from pretreatment patients. Lines 24 and 42 imply that the purpose of this study is to find predictive immune markers for HAM/TSP. However, the study using samples from patients with corticosteroid treatment can find the differences in innate immune cells between HAM/TSP and ACs but would not find prediction markers from ACs to HAM/TSP. Please modify lines 24 and 42. 

Minor points;

3．I disagree with the author's claim. They included all samples to investigate whether PVL correlates with age. However, for example, female HAM/TSP (red triangles) do not seem to correlate.

8. To know whether corticosteroids failed to control the production of inflammatory cytokines, the data that compare the production in HAM/TSP with corticoids or without those are needed.

PLOS authors have the option to publish the peer review history of their article (what does this mean?). If published, this will include your full peer review and any attached files.

Reviewer #1: No

Reviewer #2: No

Reviewer #3: No

Figure Files:

Data Requirements:

Reproducibility:

References

---

## [Editor Report · Decision Letter 2]

21 Oct 2021

Dear Dr. Hélène Dutartre,

We are pleased to inform you that your manuscript 'Immunoprofiling of fresh HAM/TSP blood samples show altered innate cell responsiveness.' has been provisionally accepted for publication in PLOS Neglected Tropical Diseases.

Best regards,

Masao Matsuoka, M.D., Ph.D.

Deputy Editor

---

## [Editor Report · Acceptance letter]

8 Nov 2021

Dear Dr Dutartre,

We are delighted to inform you that your manuscript, "Immunoprofiling of fresh HAM/TSP blood samples show altered innate cell responsiveness.," has been formally accepted for publication in PLOS Neglected Tropical Diseases.

Best regards,

Shaden Kamhawi

co-Editor-in-Chief

Paul Brindley

co-Editor-in-Chief
